# MESHMVS: MULTI-VIEW STEREO GUIDED MESH RE-CONSTRUCTION

## ABSTRACT

Deep learning based 3D shape generation methods generally utilize latent features extracted from color images to encode the objects' semantics and guide the shape generation process. These color image semantics only implicitly encode 3D information, potentially limiting the accuracy of the generated shapes. In this paper we propose a multi-view mesh generation method which incorporates geometry information in the color images explicitly by using the features from intermediate 2.5D depth representations of the input images and regularizing the 3D shapes against these depth images. Our system first predicts a coarse 3D volume from the color images by probabilistically merging voxel occupancy grids from individual views. Depth images corresponding to the multi-view color images are predicted which along with the rendered depth images of the coarse shape are used as a contrastive input whose features guide the refinement of the coarse shape through a series of graph convolution networks. Attention-based multi-view feature pooling is proposed to fuse the contrastive depth features from different viewpoints which are fed to the graph convolution networks.

We validate the proposed multi-view mesh generation method on ShapeNet, where we obtain a significant improvement with 34% decrease in chamfer distance to ground truth and 14% increase in the F1-score compared with the state-of-the-art multi-view shape generation method.

## 1 INTRODUCTION

3D shape generation is a long-standing research problem in computer vision and computer graphics with applications in autonomous driving, augmented reality, etc. Conventional approaches mainly leverage multi-view geometry based on stereo correspondences between images but are restricted by the coverage provided by the input views. With the availability of large-scale 3D shape datasets and the success of deep learning in several computer vision tasks, 3D representations such as voxel grid Choy et al. (2016); Tulsiani et al. (2017); Yan et al. (2016) and point cloud Yang et al. (2018); Fan et al. (2017) have been explored for single-view 3D reconstruction. Among them, triangle mesh representation has received the most attention as it has various desirable properties for a wide range of applications and is capable of modeling detailed geometry without high memory requirement. Single-view 3D reconstruction methods Wang et al. (2018); Huang et al. (2015); Kar et al. (2015); Su et al. (2014) generate the 3D shape from merely a single color image but suffer from occlusion and limited visibility which leads to low quality reconstructions in the unseen areas. Multi-view methods Wen et al. (2019); Choy et al. (2016); Kar et al. (2017); Gwak et al. (2017) extend the input to images from different viewpoints which provides more visual information and improves the accuracy of the generated shapes. Recent work in multi-view mesh reconstruction Wen et al. (2019) introduces a multi-view deformation network using perceptual feature from each color image for refining the meshes generated by Pixel2Mesh Wang et al. (2018). Although promising results were obtained, this method relies on perceptual features from color images which do not explicitly encode the objects' geometry and could restrict the accuracy of the 3D models.

In this work, we present a novel multi-view mesh generation method where we start by predicting coarse volumetric occupancy grid representations for the color images of each input viewpoint independently using a shared fully convolutional network which are merged into a single voxel grid in a probabilistic fashion followed by `cubify` Gkioxari et al. (2019) operation to convert it to a triangle

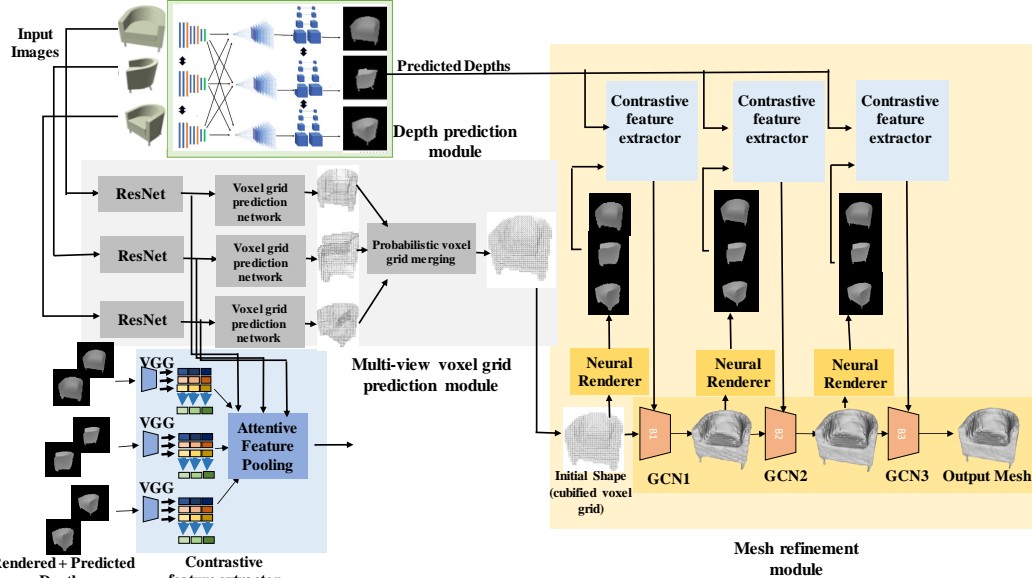

Figure 1: **Architecture of the proposed method**. The *voxel grid prediction module* predicts coarse voxel grid representation which is further refined by a series of *GCN*s. The GCNs use *contrastive depth features* from rendered depths of the current shape and the predicted depths from *MVSNet*. Multi-view features are pooled using a multi-head attention mechanism.

mesh. We then use Graph Convolutional Network (GCN) Scarselli et al. (2008); Wang et al. (2018) to fine-tune the cubified voxel grid in a coarse-to-fine manner. The GCN refines the coarse mesh by using the feature vector of each graph node (mesh vertices) obtained by projecting the vertices on the 2D contrastive depth features. The contrastive depth features are extracted from the rendered depth maps of the current mesh and predicted depth maps from a multi-view stereo network. We also propose an attention-based method to fuse feature from multiple views that can learn the importance of different views for each of the mesh vertices. Constrains between the intermediate refined mesh from GCN with predicted depth maps of different viewpoints further improve the final mesh quality. By employing multi-view voxel grid generation and refining it using geometry information from both the current mesh (through the rendered depth maps) and predicted depth maps, we are able to generate high-quality meshes. We validate our method on the ShapeNet Chang et al. (2015) benchmark and our method achieves the best performance among all previous multi-view and single-view mesh generation methods.

## 2 RELATED WORK

### 2.1 TRADITIONAL SHAPE GENERATION METHODS

3D model generation has traditionally been tackled using multi-view geometry principles. Among them, structure-from-motion (SfM) Schonberger & Frahm (2016); Agarwal et al. (2011); Cui & Tan (2015); Cui et al. (2017) and simultaneous localization and mapping (SLAM) Cadena et al. (2016); Mur-Artal et al. (2015); Engel et al. (2014); Whelan et al. (2015) are popular techniques that perform 3D reconstruction and camera pose estimation at the same time. These methods extract local image features, match them across images and use the matches to estimate camera poses and 3D geometry. Closer to our problem setup, multi-view stereo methods infer 3D geometry from images with known camera parameters. Volumetric methods Kar et al. (2017); Kutulakos & Seitz (2000); Seitz & Dyer (1999) predict voxel grid representation of objects by estimating the relationship between each voxel and object surfaces. Point cloud based methods Furukawa & Ponce (2009); Lhuillier & Quan (2005) start with a sparse point cloud and gradually increase the density of points to obtain a final dense point cloud of the object. Durou et al. (2008); Zhang et al. (1999); Favaro & Soatto (2005) reason about shading, texture and defocus to reason about visible parts of the object and infer its 3D geometry. While the results of these works are impressive in terms of quality and completeness of reconstruction,

they still struggle with poorly textured and reflective surfaces and require carefully selected input views.

## 2.2 Deep Shape Generation Methods

Deep learning based approaches can learn to infer 3D structure from training data and can be robust against poorly textured and reflective surfaces as well as limited and arbitrarily selected input views. These methods can be categorized into single view and multi-view methods. Huang et al. (2015); Su et al. (2014) use shape component retrieval and deformation from a large dataset for single-view 3D shape generation. Kurenkov et al. (2018) extend this idea by introducing free-form deformation networks on retrieved object templates from a database. Some work learn shape deformation from ground truth foreground masks of 2D images Kar et al. (2015); Yan et al. (2016); Tulsiani et al. (2017). Recurrent Neural Networks (RNN) based methods Choy et al. (2016); Kar et al. (2017); Gwak et al. (2017) are another popular solution to solve this problem. Gwak et al. (2017); Lin et al. (2019) introduce image silhouettes along with adversarial multi-view constraints and optimize object mesh models using multi-view photometric constraints. Predicting mesh directly from color images was proposed in Wang et al. (2018); Wickramasinghe et al. (2019); Pan et al. (2019); Wen et al. (2019); Gkioxari et al. (2019); Tang et al. (2019). DR-KFS Jin et al. (2019) introduces a differentiable visual similarity metric while SeqXY2SeqZ Han et al. (2020) represents 3D shapes using a set of 2D voxel tubes for shape reconstruction. Front2Back Yao et al. (2020) generates 3D shapes by fusing predicted depth and normal images and DV-Net Jia et al. (2020) predicts dense object point clouds using dual-view RGB images with a gated control network to fuse point clouds from the two views. FoldingNet Yang et al. (2018) learns to reconstruct arbitrary point clouds from a single 2D grid. AtlasNet Groueix et al. (2018) use learned parametric representation while Mescheder et al. (2019); Park et al. (2019); Liu et al. (2019b;a); Murez et al. (2020) employ implicit surface representation to reconstruct 3D shapes.

## 2.3 Depth Estimation

Compared to 3D shape generation, depth prediction is an easier problem formulation since it simplifies the task to per-view depth map estimation. Traditional methods Campbell et al. (2008); Galliani et al. (2015); Schönberger et al. (2016) use multi-view stereo principles for depth prediction. Deep learning based multi-view stereo depth estimation was first introduced in Hartmann et al. (2017) where a learned cost metric is used to estimate patch similarities. DeepMVS Huang et al. (2018) warps multi-view images to 3D space and then applies deep networks for regularization and aggregation to estimate depth images. Learned 3D cost volume based depth prediction was proposed in MVSNet Yao et al. (2018) where a 3 dimensional cost volume is built using homographically warped 2D features from multi-view images and 3D CNNs are used for cost regularization and depth regression. This idea was further extended by Chen et al. (2019); Luo et al. (2019); Gu et al. (2019); Yao et al. (2019).

## 3 Methodology

Figure 1 shows the architecture of the proposed system which takes as input multi-view color images of an object with known poses and outputs a triangle mesh representing the surface of the object.

### 3.1 Multi-view Voxel Grid Prediction

**Single-view Voxel Grid Prediction**   The single-view voxel branch consists of a ResNet feature extractor and a fully convolutional voxel grid prediction network. It generates the coarse initial shape of an object from one viewpoint as voxel occupancy grid using a color image. Here, we set the resolution of the generated voxel occupancy grid as $32 \times 32 \times 32$. The voxel prediction networks for all viewpoints share the same weights.

**Probabilistic Occupancy Grid Merging**   Voxel occupancy grid predicted from a single viewpoint suffers from occlusion and limited visibility. In order to fuse voxel grids from different viewpoints, we propose a probabilistic occupancy grid merging method which merges the voxel grids from each input viewpoint probabilistically to obtain the final voxel grid output. This allows occluded regions in one view to be estimated from other views where those regions are visible as well as increase the

confidence of prediction in overlapping regions. Occupancy probability of each voxel is represented by $p(x)$ which is converted to log-odds (logit):

$$l(x) = log\frac{p(x)}{1 - p(x)} \tag{1}$$

Bayesian update on the probabilities reduce to simple summation of log likelihoods Konolige (1997). Hence, the multi-view log-odds of a voxel is given by:

$$l(x) = l_1(x) + l_2(x) + ... + l_n(x) \tag{2}$$

where $l_i$ is the voxel's log-odds in view $i$ and $n$ is the number of input views. The final voxel probability $x$ is obtained by applying the inverse function of Equation (1) which is a sigmoid function.

## 3.2 MESH REFINEMENT

The `cubified` mesh from the voxel branch only provides a coarse reconstruction of the object's surface. We apply graph convolutional networks which represent each mesh vertex as one graph node and deforms them to more accurate positions.

**GCN-based Mesh Deformation**   The features pooled from multi-view images along with 3D coordinates of the vertices in world frame are used as features of the graph nodes. Series of Graph-based Convolutional Network (GCN) blocks are applied to deform a mesh at the current stage to the next stage, starting with the `cubified` voxel grids. A graph convolution deforms mesh vertices by propagating features from neighboring vertices by applying $f_i^{'} = ReLU(W_0 f_i + \sum_{j \in \mathcal{N}(i)} W_1 f_j)$ where $\mathcal{N}(i)$ is the set of neighboring vertices of the $i$-th vertex in the mesh, $f_{\{\}}$ represents the feature vector of a vertex, and $W_0$ and $W_1$ are learnable parameters of the model. Each GCN block utilizes several graph convolutions to transform the vertex features along with a final *vertex refinement* operation where the features along with vertex coordinates are further transformed as $v_i^{'} = v_i + tanh(W_{vert}[f_i; v_i])$ where the matrix $W_{vert}$ is another learnable parameter to obtain the deformed mesh.

**Contrastive Depth Feature Extraction**   Yao et al. (2020) demonstrate that using intermediate, image-centric 2.5D representations instead of directly generating 3D shapes in global frame from raw 2D images can improve 3D reconstruction quality. We therefore propose to formulate the features for graph nodes using 2.5D depth maps as input additional inputs alongside the RGB features. Specifically, we render the meshes at different GCN stages to depth image at all the input views using Kato et al. (2018) and use them along with predicted depths for depth feature extraction. We call this form of depth input `contrastive depth` as it contrasts the rendered depths of the current mesh against the predicted depths and allows the network to reason about the deformation better than when using predicted depth or color images alone. Given the 2D features, corresponding feature vectors of individual vertices can be found by projecting the 3D vertex coordinates to the feature planes using known camera parameters. We use VGG-16 Simonyan & Zisserman (2014) as our contrastive depth feature extraction network.

**Multi-View Depth Estimation**   We extend MVSNet Yao et al. (2018) and predict the depth maps of all views since the original implementation predicts depth of only one reference view. This is achieved by transforming the feature volumes to each view's coordinate frame using homography warping and applying identical cost volume regularization and depth regression on each view. Detailed network architecture diagram of this module is provided in the appendix.

**Attention-based Multi-View Feature Pooling**   In order to fuse multi-view contrastive depth features, we formulate an attention module by adapting multi-head attention mechanism originally designed for sequence to sequence machine translation using transformer (encoder-decoder) architecture Vaswani et al. (2017). In a transformer architecture the encoder hidden state is mapped to lower dimension key-value pairs ($\mathbf{K}$, $\mathbf{V}$) while the decoder hidden state is mapped to a query vector $\mathbf{Q}$ using independent fully connected layers. The encoder hidden state in our case is the multi-view features while the decoder hidden state is the *mean* of the multi-view features. The attention weights are computed using scaled-dot product:

$$Attention(\mathbf{Q}, \mathbf{K}, \mathbf{V}) = softmax(\frac{\mathbf{QK}^T}{\sqrt{N}})\mathbf{V} \tag{3}$$

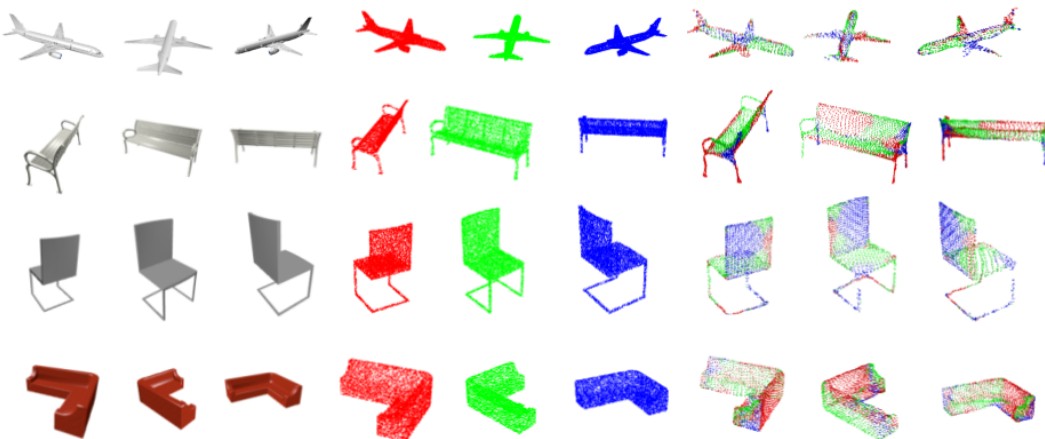

Figure 2: **Attention weights visualization.** From left to right: input images from 3 viewpoints, corresponding ground truth point clouds color-coded by their view order and the predicted mesh vertices color-coded by the attention weights of the views. Only the view with maximum attention weight is visualized for each predicted points for clarity.

where $N$ is the number of input views.

Multiple attention *heads* are used which are concatenated and transformed to obtain the final output

$$head_i = Attention(\mathbf{Q}\mathbf{W}_i^Q, \mathbf{K}\mathbf{W}_i^K, \mathbf{V}\mathbf{W}_i^V) \qquad (4)$$

$$MultiHead(\mathbf{Q}, \mathbf{K}, \mathbf{V}) = [head_1; ...; head_h]\mathbf{W}^0 \qquad (5)$$

where multiple $\mathbf{W}$ are parameters to be learned, $h$ is the number of attention heads and $i \in [1, h]$.

We choose multi-head attention as our feature pooling method since it allows the model to attend information from different representation subspaces of the features by training multiple attentions in parallel. This method is also invariant to the order and number of input views. We visualize the learned attention weights (average of each attention heads) in Figure 2 where we can observe that the attention weights roughly takes into account the visibility/occlusion information from each view.

### 3.3 LOSS FUNCTIONS

**Mesh losses** The losses which are derived from Wang et al. (2018) to constrain the mesh predicted by each GCN block (P) to resemble the ground truth (Q) include Chamfer distance $\mathcal{L}_{\text{chamfer}}(P, Q) = |P|^{-1} \sum_{(p,q) \in \Lambda_{P,Q}} ||p - q||^2 + |Q|^{-1} \sum_{(q,p) \in \Lambda_{Q,P}} ||q - p||^2$ and surface normal loss $\mathcal{L}_{\text{normal}}(P, Q) = -|P|^{-1} \sum_{(p,q) \in \Lambda_{P,Q}} |u_p \cdot u_q| - |Q|^{-1} \sum_{(q,p) \in \Lambda_{Q,P}} |u_q \cdot u_p|$ with additional regularization in the form of edge length loss $\mathcal{L}_{\text{edge}}(V, E) = \frac{1}{|E|} \sum_{(v,v') \in E} ||v - v'||^2$ for visually appealing results.

**Depth loss** Our depth prediction network is supervised using adaptive reversed Huber loss (also known as BerHu criterion) Lambert-Lacroix & Zwald (2016). $\mathcal{L}_{depth} = |x|, \text{if } |x| \leq c$, otherwise $\frac{x^2+c^2}{2c}$ where $x$ is the depth error of a pixel and $c$ is a constant set to $0.2$. Note that the original MVSNet uses L1-loss, but we used BerHu loss since it gave slightly higher accuracy. Intuitively, this is because BerHu provides a good balance between L1 and L2 loss and has shown similar improvement in Laina et al. (2016).

**Contrastive depth loss** BerHu loss is also applied between the rendered depth images at different GCN stages and the predicted depth images. $\mathcal{L}_{contrastive} = |x|, \text{if } |x| \leq c$, otherwise $\frac{x^2+c^2}{2c}$

**Voxel loss** Binary cross-entropy loss between the predicted voxel occupancy probabilities and the ground truth occupancies is used as voxel loss to supervise the voxel predictions $\mathcal{L}_{\text{voxel}} = -\Big(p(x)log\big(p(x)\big) + \big(1 - p(x)\big)log\big(1 - p(x)\big)\Big)$

**Final loss** We use the weighted sum of the individual losses discussed above as the final loss to train our model in an end-to-end fashion. $\mathcal{L} = \lambda_{\text{chamfer}}\mathcal{L}_{\text{chamfer}} + \lambda_{\text{normal}}\mathcal{L}_{\text{normal}} + \lambda_{\text{edge}}\mathcal{L}_{\text{edge}} + \lambda_{\text{depth}}\mathcal{L}_{\text{depth}} + \lambda_{\text{contrastive}}\mathcal{L}_{\text{contrastive}} + \lambda_{\text{voxel}}\mathcal{L}_{\text{voxel}}$ , where $\mathcal{L}$ is the final loss term.

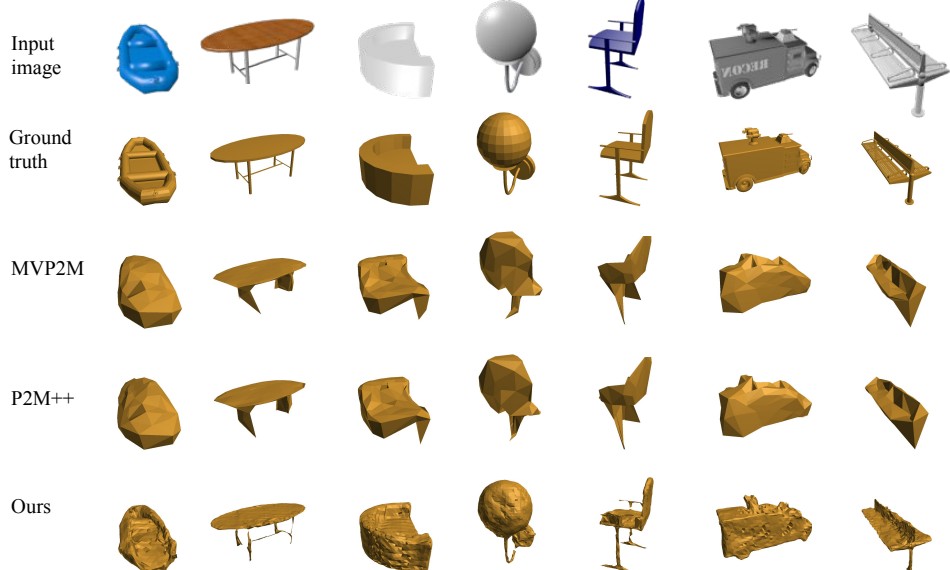

Figure 3: **Qualitative evaluation** on ShapeNet dataset. **From top to bottom**: one of the input images, ground truth mesh, multi-view extended Pixel2Mesh, Pixel2Mesh++, and ours. Our predictions are closer to the actual shape, especially for the objects with more complex topologies.

# 4 EXPERIMENTS

## 4.1 EXPERIMENTAL SETUP

**Comparisons**   We evaluate the proposed method against various multi-view shape generation methods. The state-of-the-art method is Pixel2Mesh++ Wen et al. (2019) (referred as *P2M++*). Wen et al. (2019) also provide a baseline by directly extending Pixel2Mesh Wang et al. (2018) to operate on multi-view images (referred as *MVP2M*) using their statistical feature pooling method to aggregate features from multiple color images. Results from additional multi-view shape generation baselines 3D-R2N2 Choy et al. (2016) and LSM Kar et al. (2017) are also reported.

**Dataset**   We evaluate our method against the state-of-the-art methods on the dataset from Choy et al. (2016) which is a subset of ShapeNet Chang et al. (2015) and has been widely used by recent 3D shape generation methods. It contains 50K 3D CAD models from 13 categories. Each model is rendered with a transparent background from 24 randomly chosen camera viewpoints to obtain color images. The corresponding camera intrinsics and extrinsics are provided in the dataset. Since the dataset does not contain depth images, we render them using a custom depth renderer at the same viewpoints as the color images and with the same camera intrinsics. We follow the training/testing/validation split of Gkioxari et al. (2019).

**Implementation**   For the depth prediction module, we follow the original MVSNet Yao et al. (2018) implementation. The output depth dimensions reduces by a factor of 4 to $56{\times}56$ from the $224{\times}224$ input image. The number of depth hypotheses is chosen as 48 which offers a balance between accuracy and running/training time efficiency. These depth hypotheses represent values from $0.1$ m to $1.3$ m at an interval of $25$ mm. These values were chosen based on the range of depths present in the dataset.

The hierarchical features obtained from "Contrastive Depth Features Extractor" are of total 4800 dimensions for each view. The aggregated multi-view features are compressed to 480 dimensional after applying attentive feature pooling. 5 attention heads are used for merging multi-view features. The loss function weights are set as $\lambda_{\text{chamfer}} = 1$, $\lambda_{\text{normal}} = 1.6 \times 10^{-4}$, $\lambda_{\text{depth}} = 0.1$, $\lambda_{\text{contrastive}} = 0.001$ and $\lambda_{\text{voxel}} = 1$. Two settings of $\lambda_{\text{edge}}$ were used, $\lambda_{\text{edge}} = 0$ (referred as *Best*) which gives better quantitative results and $\lambda_{\text{edge}} = 0.2$ (referred as *Pretty*) which gives better qualitative results.

**Training and Runtime**   The network is optimized using Adam optimizer with a learning rate of $10^{-4}$. The training is done on 5 Nvidia RTX-2080 GPUs with effective batch size 5. The depth prediction network (MVSNet) is trained independently for 30 epochs. Then the whole system is

trained for another 40 epochs with the weights of the MVSNet frozen. Our system is implemented in PyTorch deep learning framework and it takes around 60 hours for training.

**Evaluation Metric**   Following Wang et al. (2018); Wen et al. (2019), we use F1-score as our evaluation metric. The F1-score is the harmonic mean of precision and recall where the precision/recall are calculated by finding the percentage of points in the predicted/ground truth that can find a nearest neighbor from the other within a threshold. We provide evaluations with two threshold values: $\tau$ and $2\tau$ where $\tau = 10^{-4}$ m$^2$.

## 4.2   COMPARISON WITH PREVIOUS MULTI-VIEW SHAPE GENERATION METHODS

We quantitatively compare our method against previous works for multi-view shape generation in Table 1 and show the effectiveness of our methods in improving the shape quality. Our method outperforms the state-of-the-art method Pixel2Mesh++ Wen et al. (2019) with a decrease in chamfer distance to ground truth by 34% and 15% increase in F1-score at threshold $\tau$. Note that in Table 1 the same model is trained for all the categories but accuracy on individual categories as well as average over the categories are evaluated. We provide the chamfer distances in the appendix.

| Category | F-score ($\tau$) ↑ | | | | | | F-score ($2\tau$) ↑ | | | | | |
|---|---|---|---|---|---|---|---|---|---|---|---|---|
| | 3D-R2N2 | LSM | MVP2M | P2M++ | Ours (pretty) | **Ours (best)** | 3D-R2N2 | LSM | MVP2M | P2M++ | Ours (pretty) | **Ours (best)** |
| Couch | 45.47 | 43.02 | 53.17 | 57.56 | 71.63 | **73.63** | 59.97 | 55.49 | 73.24 | 75.33 | 85.28 | **88.24** |
| Cabinet | 54.08 | 50.80 | 56.85 | 65.72 | 75.91 | **76.39** | 64.42 | 60.72 | 76.58 | 81.57 | 87.61 | **88.84** |
| Bench | 44.56 | 49.33 | 60.37 | 66.24 | 81.11 | **83.76** | 62.47 | 65.92 | 75.69 | 79.67 | 90.56 | **92.57** |
| Chair | 37.62 | 48.55 | 54.19 | 62.05 | 77.63 | **78.69** | 54.26 | 64.95 | 72.36 | 77.68 | 88.24 | **90.02** |
| Monitor | 36.33 | 43.65 | 53.41 | 60.00 | 74.14 | **76.64** | 48.65 | 56.33 | 70.63 | 75.42 | 86.04 | **88.89** |
| Firearm | 55.72 | 56.14 | 79.67 | 80.74 | 92.92 | **94.32** | 76.79 | 73.89 | 89.08 | 89.29 | 96.81 | **97.67** |
| Speaker | 41.48 | 45.21 | 48.90 | 54.88 | 66.02 | **67.83** | 52.29 | 56.65 | 68.29 | 71.46 | 79.76 | **82.34** |
| Lamp | 32.25 | 45.58 | 50.82 | 62.56 | 75.91 | **75.93** | 49.38 | 64.76 | 65.72 | 74.00 | 82.00 | **85.33** |
| Cellphone | 58.09 | 60.11 | 66.07 | 74.36 | 85.57 | **86.45** | 69.66 | 71.39 | 82.31 | 86.16 | 93.40 | **94.28** |
| Plane | 47.81 | 55.60 | 75.16 | 76.79 | 89.23 | **92.13** | 70.49 | 76.39 | 86.38 | 86.62 | 94.65 | **96.57** |
| Table | 48.78 | 48.61 | 65.95 | 71.89 | 82.37 | **83.68** | 62.67 | 62.22 | 79.96 | 84.19 | 90.24 | **91.97** |
| Car | 59.86 | 51.91 | 67.27 | 68.45 | 77.01 | **80.43** | 78.31 | 68.20 | 84.64 | 85.19 | 88.99 | **92.33** |
| Watercraft | 40.72 | 47.96 | 61.85 | 62.99 | 75.52 | **80.48** | 63.59 | 66.95 | 77.49 | 77.32 | 86.77 | **90.35** |
| Mean | 46.37 | 49.73 | 61.05 | 66.48 | 78.58 | **80.80** | 62.53 | 64.91 | 77.10 | 80.30 | 88.49 | **90.72** |

Table 1: **Qualitative comparison** against state-of-the-art multi-view shape generation methods. We report F-score on each semantic category along with the mean over all categories using two thresholds $\tau$ and $2\tau$ for nearest neighbor match where $\tau=10^{-4}$ m$^2$.

We also provide visual results for qualitative assessment of the generated shapes by our *Pretty* model in Figure 3 which shows that it is able to more accurately predict topologically diverse shapes.

## 4.3   ABLATION STUDIES

**Contrastive Depth Feature Extraction**   We evaluate several methods for contrastive feature extraction (Sub-section 3.2). These methods are 1) *Input Concatenation*: using the concatenated rendered and predicted depth maps as input to the VGG feature extractor, 2) *Input Difference*: using the difference of the two depth maps as input to VGG, 3) *Feature Concatenation*: concatenating features from rendered and predicted depths extracted by shared VGG, 4) *Feature Difference*: using difference of the features from the two depth maps extracted by shared VGG, and 5) *Predicted depth only*: using the VGG features from the predicted depths only. 6) *Rendered depth only*: using the VGG features from the rendered depths only. The quantitative results are summarized in Table 2 and shows that *Input Concatenation* method produces better results than other formulations.

**Accuracy with different settings**   Table 3 shows the contribution of different components towards the final accuracy. Naively extending the single-view Mesh R-CNN Gkioxari et al. (2019) to multiple views using statistical feature pooling Wen et al. (2019) for mesh refinement (row 1) gives an F1-score of 72.74% for threshold $\tau$ which is 6.26% improvement over Pixel2Mesh++. We further extend the above method with our probabilistic multi-view voxel grid prediction in row 2 and get a 4.23% improvement.

In row 3 of Table 3 we use our contrastive depth features instead of RGB features for mesh refinement and get 2.7% improvement. We then replace the statistical feature pooling with the proposed attention method and get 0.19% improvement. The improvement is not significant on our final architecture but we found the multi-head attention to perform better on more light-weight architectures. We also evaluate the effect of using additional regularization from contrastive depth losses: rendered depth vs predicted depth in the 5th rows of which improves the score by 0.98%. In row 6 we use ground truth

|  | F1-$\tau$ | F1-2$\tau$ |
|---|---|---|
| (1) Input Concatenation | **80.80** | **90.72** |
| (2) Input Difference | 80.41 | 90.54 |
| (3) Feature Concatenation | 80.45 | 90.54 |
| (4) Feature Difference | 80.30 | 90.40 |
| (5) Predicted Depth only | 79.40 | 89.95 |
| (6) Rendered Depth only | 78.20 | 88.90 |

Table 2: **Comparisons of different contrastive depth formulations**. In 1st and 2nd rows, concatenation and difference of the rendered and predicted depths are fed to VGG feature extractor while in 3rd and 4th rows, concatenation and difference of the VGG features from the depths is used for mesh refinement. 5 uses VGG features from predicted depths only while 6 uses VGG features from rendered depths only.

instead of predicted depths on our final model which gives the upper bound on our mesh prediction accuracy in relation to the depth prediction accuracy as 84.58%.

|  | F1-$\tau$ | F1-2$\tau$ |
|---|---|---|
| (1) Naive multi-view Mesh R-CNN | 72.74 | 84.99 |
| (2) + Multi-view voxel grid prediction | 76.97 | 88.24 |
| (3) + Contrastive depth input | 79.63 | 90.10 |
| (4) + Multi-head attention pooling | 79.82 | 90.18 |
| (5) **+ Contrastive depth loss (final model)** | **80.80** | **90.72** |
| (6) Using GT depth (final model) | **84.58** | **92.86** |

Table 3: **Comparison of shape generation accuracy with different settings** of additional contrastive depth losses, multi-view feature pooling. The Baseline framework uses multi-head attention mechanism without any contrastive depth losses.

**Number of View** We test the performance of our framework with respect to the number of views. Table 4 shows that the accuracy of our method increases as we increase the number of input views for training. These experiments also validate that the attention-based feature pooling can efficiently encode features from different views to take advantage of larger number of views.

Table 5 shows the results when using different number of views during testing on our model trained with 3 views which indicates that increasing the number of views during testing does not improve the accuracy while decreasing the number of views can cause a significant drop in accuracy.

| Metric | 2 | 3 | 4 | 5 | 6 |
|---|---|---|---|---|---|
| F1-$\tau$ | 73.60 | 80.80 | 82.61 | 83.76 | 84.25 |
| F1-2$\tau$ | 85.80 | 90.72 | 91.78 | 92.73 | 93.14 |

| Metric | 2 | 3 | 4 | 5 | 6 |
|---|---|---|---|---|---|
| F1-$\tau$ | 72.46 | 80.80 | 80.98 | 80.94 | 80.85 |
| F1-2$\tau$ | 84.49 | 90.72 | 91.03 | 91.16 | 91.20 |

Table 4: **Accuracy w.r.t the number of views during training**. The evaluation was performed on the same number of views as training.

Table 5: **Accuracy w.r.t the number of views during testing**. The same model trained with 3 views was used in all of the cases.

## 5 CONCLUSION

We propose a neural network based solution to predict 3D triangle mesh models of objects from images taken from multiple views. First, we propose a multi-view voxel grid prediction module which probabilistically merges voxel grids predicted from individual input views. We then cubify the merged voxel grid to triangle mesh and apply graph convolutional networks for further refining the mesh. The features for the mesh vertices are extracted from contrastive depth input consisting of rendered depths at each refinement stage along with the predicted depths. The proposed mesh reconstruction method outperforms existing methods with a large margin and is capable of reconstructing objects with more complex topologies.

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

# A  APPENDIX

## NETWORK ARCHITECTURE

### MVSNET ARCHITECTURE

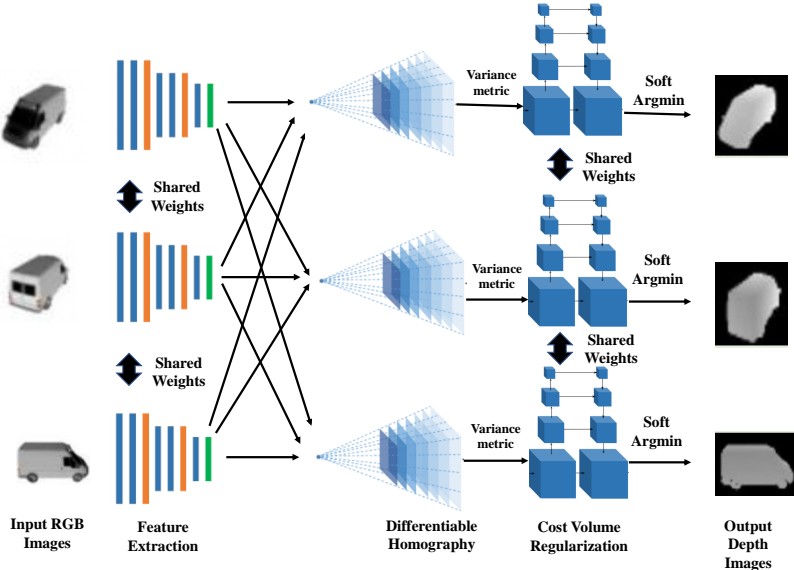

Figure 4: Depth prediction network (MVSNet) architecture

Our depth prediction module is based on MVSNet Yao et al. (2018) which constructs a regularized 3D cost volumes to estimate the depth map of the reference view. Here, we extent MVSNet to predict the depth maps of all views instead of only the reference view. This is achieved by transforming the feature volumes to each view's coordinate frame using homography warping and applying identical cost volume regularization and depth regression on each view. This allows the reuse of pre-regularization feature volumes for efficient multi-view depth prediction invariant to the order of input images. Figure 4 shows the architecture of the our depth estimation module.

PROBABILISTIC OCCUPANCY GRID MERGING

We use single-view voxel prediction network from Gkioxari et al. (2019) to predict predicts voxel grids for each of the input images in their respective local coordinate frames. The occupancy grids are transformed to global frame (which is set to the coordinate frame of the first image) by finding the equivalent global grid values in the local grids after applying bilinear interpolation on the closest matches. The voxel grids in global coordinates are then probabilistically merged according to Sub-section 3.1 of the main submission.

EXPERIMENTS

We quantitatively compare our method against previous works for multi-view shape generation in Table 6 and show effectiveness of our proposed shape generation methods in improving shape quality. Our method outperforms the state-of-the-art method Pixel2Mesh++ Wen et al. (2019) with decrease in chamfer distance to ground truth by 34%, which shows the effectiveness of our proposed method. Note that in Table 6 same model is trained for all the categories but accuracy on individual categories as well as average over all the categories are evaluated.

| Category | Chamfer Distance (CD) $\downarrow$ | | | | |
| | 3D-R2N2 | LSM | MVP2M | P2M++ | **Ours** |
|---|---|---|---|---|---|
| Couch | 0.806 | 0.730 | 0.534 | 0.439 | **0.220** |
| Cabinet | 0.613 | 0.634 | 0.488 | 0.337 | **0.230** |
| Bench | 1.362 | 0.572 | 0.591 | 0.549 | **0.159** |
| Chair | 1.534 | 0.495 | 0.583 | 0.461 | **0.201** |
| Monitor | 1.465 | 0.592 | 0.658 | 0.566 | **0.217** |
| Firearm | 0.432 | 0.385 | 0.305 | 0.305 | **0.123** |
| Speaker | 1.443 | 0.767 | 0.745 | 0.635 | **0.402** |
| Lamp | 6.780 | 1.768 | 0.980 | 1.135 | **0.755** |
| Cellphone | 1.161 | 0.362 | 0.445 | 0.325 | **0.138** |
| Plane | 0.854 | 0.496 | 0.403 | 0.422 | **0.084** |
| Table | 1.243 | 0.994 | 0.511 | 0.388 | **0.181** |
| Car | 0.358 | 0.326 | 0.321 | 0.249 | **0.165** |
| Watercraft | 0.869 | 0.509 | 0.463 | 0.508 | **0.175** |
| Mean | 1.455 | 0.664 | 0.541 | 0.486 | **0.211** |

Table 6: **Qualitative comparison** against state-of-the-art multi-view shape generation methods. Following Wen et al. (2019), we report Chamfer Distance in $m^2 \times 1000$ from ground truth for different methods. Note that same model is trained for all the categories but accuracy on individual categories as well as average over all the categories are evaluated.

ABLATION STUDIES

**Coarse Shape Generation** We conduct comparisons on voxel grid predicted from our proposed probabilistically merged voxel grids against single view method Gkioxari et al. (2019). As is shown in Table 7, the accuracy of the initial shape generated from probabilistically merged voxel grid is higher than that from individual views.

**Accuracy at Different GCN Stages** We analyze the accuracy of meshes at different GCN stages in Table 8. The results validate that our method produces the meshes in a coarse-to-fine manner and multiple GCN refinements improve the mesh quality.

**Resolution of Depth Prediction** We conduct experiments using different numbers of depth hypotheses in our depth prediction network (Sub-section A), producing depth values at different resolutions. A higher number of depth hypothesis means finer resolution of the predicted depths. The quantitative results with different hypothesis numbers are summarized in Table 9. We set depth hypothesis as 48 for our final architecture which is equivalent to the resolution of 25 mm. We observe that the mesh accuracy remain relatively unchanged if we predict depths at finer resolutions.

| Metric | Single-view | Multi-view |
|--------|-------------|------------|
| F1-$\tau$ | 25.19 | 31.27 |
| F1-2$\tau$ | 36.75 | 44.46 |

| Metric | Cubified | Stage-1 | Stage-2 | Stage-3 |
|--------|----------|---------|---------|---------|
| F1-$\tau$ | 31.48 | 76.78 | 79.88 | 80.80 |
| F1-2$\tau$ | 44.40 | 88.32 | 90.19 | 90.72 |

Table 7: **Accuracy of predicted voxel grids** from single-view prediction compared against the proposed probabilistically merged multi-view voxel grids. The voxel branch was trained separately without the mesh refinement and evaluation was performed on the cubified voxel grids. We use three views for probabilistic grid merging.

Table 8: **Accuracy of the refined meshes at different GCN stages**. 1, 2 and 3 indicate the performance at the corresponding graph convolution blocks while *Cubified* is for the cubified voxel grids used as input for the first GCN block. All the stages, including the voxel prediction, were trained jointly and hence the accuracy of voxel predictions varies from that in Table 7.

| Metric | 24 | 48 | 72 | 96 |
|--------|------|------|------|------|
| F1-$\tau$ | 80.29 | 80.80 | 80.69 | 80.34 |
| F1-2$\tau$ | 90.43 | 90.72 | 90.74 | 90.47 |

Table 9: **Accuracy w.r.t the number of depth hypothesis**. A higher number of depth hypothesis increases the resolution of predicted depth values at the expense of higher memory requirement. The range of depths for all the models are same and based on the minimum/maximum depth in the ShapeNet Chang et al. (2015) dataset.

**Generalization Capability** We conduct experiments to evaluate the generalization capability of our system across the semantic categories. We train our model with only 12 out of the 13 categories and test on the category that was left out. Table 10 shows that the accuracy generally does not decrease significantly when compared with the model that was trained on all 13 categories when using 2$\tau$ threshold for the F-score.

| Category | F-score ($\tau$) ↑ | | F-score ($2\tau$) ↑ | |
|----------|-----------|-----------|-----------|-----------|
| | Excluding | Including | Excluding | Including |
| Couch | 63.29 | 73.63 | 80.79 | 88.24 |
| Cabinet | 68.26 | 76.39 | 83.10 | 88.84 |
| Bench | 76.08 | 83.76 | 87.42 | 92.57 |
| Chair | 60.60 | 78.69 | 75.93 | 90.02 |
| Monitor | 67.26 | 76.64 | 81.57 | 88.89 |
| Firearm | 78.59 | 94.32 | 86.28 | 97.67 |
| Speaker | 62.39 | 67.83 | 77.77 | 82.34 |
| Lamp | 63.50 | 75.93 | 74.66 | 85.33 |
| Cellphone | 67.24 | 86.45 | 80.54 | 94.28 |
| Plane | 57.48 | 92.13 | 67.27 | 96.57 |
| Table | 76.41 | 83.68 | 86.86 | 91.97 |
| Car | 59.08 | 80.43 | 75.58 | 92.33 |
| Watercraft | 64.97 | 80.48 | 78.95 | 90.35 |

Table 10: **Accuracy when a category is excluded** during training and evaluation is performed on the category to verify how well training on other categories generalizes to the excluded category.

# B APPENDIX

## BEST VS PRETTY MODELS

We provide qualitative comparison between the our models trained with *best* and *pretty* configurations in Figure 5. The *best* configuration refers to our model trained without edge regularization while *pretty* refers to the model trained with the regularization (Sub-section 4.1). We observe that without the regularization we get higher score on our evaluation metrics but get degenerate meshes with self-intersections and irregularly sized faces.

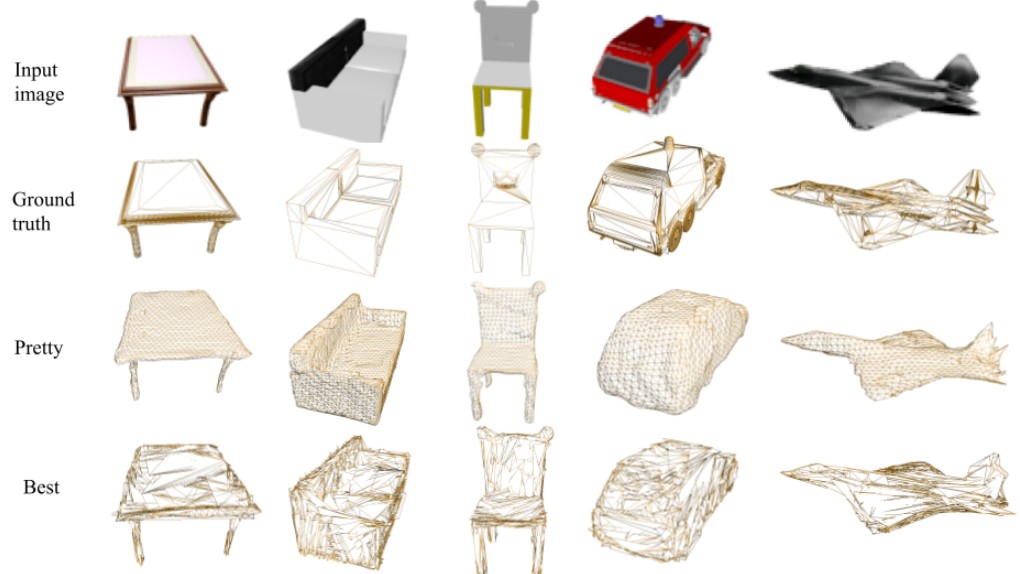

Figure 5: Qualitative evaluation: best vs pretty wireframe models. The best models while being preferred by the evaluation metrics lead to degenerate meshes, with irregularly sized faces and self-intersections

## FAILURE CASES

Some failure cases of our model (with pretty setting) are shown in Figure 6. We notice that the rough topology of the mesh is recovered while we failed to reconstruct the fine topology. We can regard the recovery from wrong initial topology as a promising future work.

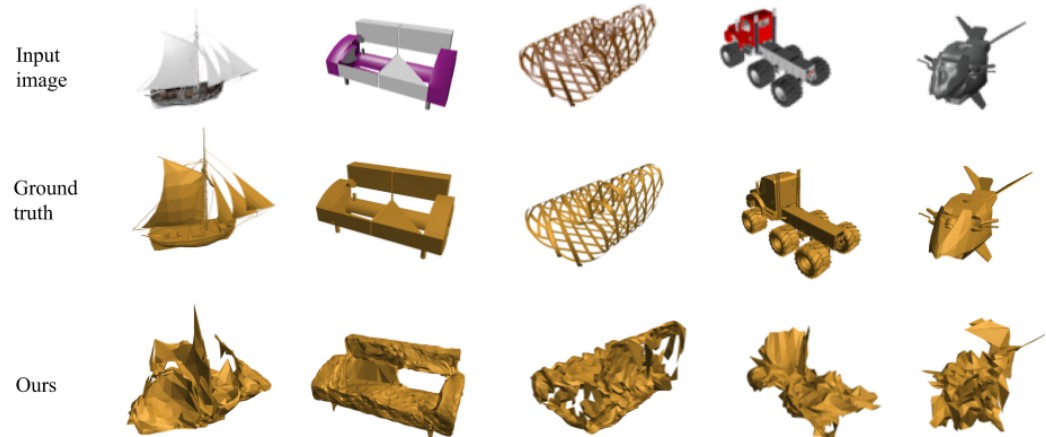

Figure 6: Failure Cases. Our system can struggle to roughly reconstruct shapes with very complex topology while some fine topology of the mesh is missing.

