# OpenReview forum: "MeshMVS: Multi-view Stereo Guided Mesh Reconstruction"
_ICLR.cc/2021/Conference — Reject_

### Official Review · AnonReviewer1 · 2020-10-27
**Solid work on multi-view mesh reconstruction**

**Rating:** 9
**Confidence:** 3

**Review:**

**Quality:**
Overall the quality of this work is high.  The quantitative and qualitative results are impressive relative to the SoA. I would like to see the qualitative results for the Best model as opposed to just the Pretty model, and I'm curious why the best qualitative mode was not the same as the best quantitative model.  I would think analyzing this difference could give the authors insight into how to improve the model.

**Clarity:**
Overall the paper is written clearly, explaining and justifying the different components of the model clearly.  There are a few issues/questions I have:

* Page 2: change "non-reflective" -> "reflective"
* For depth estimation, I'm wondering why you changed the MVSNet loss function to use BerHu instead of L1 used in the original paper?
* Could you define the terms in the BerHu criterion?  What are x and c? It would also be good to shed some intuition on why this criterion is the right one.
* The mixing constants in your loss function ($\lambda$) vary across several orders of magnitude.  How were those selected?
* On page 6 you state that two values of $\tau$ are used, but elsewhere in the paper $\tau$ is defined as $10^{-4}$ and you use $\tau$ and $2\tau$.

**Originality:**
The paper generally uses a mix of SoA techniques creatively woven together in a fairly sophisticated model. Oher novel aspects such as using the neural renderer to create the contrastive depth module was interesting.


**Significance:**
This work is significant based on the importance of the problem - this is one of the harder and most important problems in computer vision today,  in the quality of its results and in the creative way it combines SoA methods to provide multiple semi-supervised losses.

---

> ### Author Response · Authors · 2020-11-12
> **Response to AnonReviewer1**
>
> We really appreciate your insightful comments on our manuscript. We will revise the typos in the updated version.
>
> **Q1: Why the best qualitative mode was not the same as the best quantitative model?**
>
> A: Yes, Mesh R-CNN and Pixel2Mesh also observe that the metrics for shape reconstruction do not always correlate well with the shapes' visual quality. It is mainly because training without shape regularizers give meshes that are preferred by metrics despite being highly degenerate, with irregularly-sized faces and many self-intersections.
>
> **Q2: Why you changed the MVSNet loss function to use BerHu instead of L1 used in the original paper? What are x and c? It would also be good to shed some intuition on why this criterion is the right one**
>
> A: We found our modified MVS-Net converges to a better accuracy with BerHu loss than with L1 loss.
> In the BerHu criterion $\mathcal{L} = |x|, if |x| \le c,\text{ otherwise }\frac{x^2 + c^2}{2c}$, x is the depth error of a pixel and $c$ is a constant threshold that is set to $0.2$ in our implementation.
>
> Intuitively, BerHu criterion acts as L1 loss for smaller errors and L2 loss for larger ones. The constant $c$ determines where the switch happens. BerHu criterion weights larger residuals more heavily while at the same time having higher gradient for the smaller residuals. This has been shown to work better for predicting depth values which usually follow a heavy-tailed distribution.
>
> BA-Net and ``Deeper Depth Prediction with Fully Convolutional Residual Networks'' also use BerHu loss for depth prediction for this reason.
>
> **Q3: The mixing constants in your loss function ($\lambda$) vary across several orders of magnitude. How were those selected?**
>
> A: The constants for the loss terms chamfer loss, normal loss, edge loss and voxel loss are from Mesh R-CNN. The constants for the remaining loss terms, viz., contrastive and predicted depth losses were obtained by hyperparameter tuning.
>
> **Q4: I would like to see the qualitative results for the Best model as opposed to just the Pretty mode.**
>
> A: Of course, we will provide these qualitative results in the Appendix B of our revised manuscript.

---

### Official Review · AnonReviewer2 · 2020-10-28
**Positive towards the results, conservative towards the contribution**

**Rating:** 6
**Confidence:** 3

**Review:**

Overview

This paper proposes a system of reconstructing 3D objects from multi-view images. The system consists of a single-view voxel generation network, a multi-view voxel fusion mechanism, a multi-view depth estimation network, and a refinement network aggregating multi-view depth features.
The major contribution is in the refinement stage upon the coarse reconstruction obtained from voxel predictions, typically for the introduction of the Attention-based Multi-View Feature Pooling.

Method Novelty
According to the paper and the attached code, it seems like the authors mostly utilized existing networks to build a system. The author introduces their Attention-based Multi-View Feature Pooling mechanism which is new. Despite the results, the system is rather bulky and ad-hoc. For the use of GCN in refinement, see Question 2.

Results
The paper achieves plausible state-of-the-arts quantitate results on standard evaluation sets and metrics.
The visual quality is reasonable, however from Figure 3 it seems like reconstructed local surface suffers from noises. Their results struggles to getting clean surface especially when compared to implicit-based methods, such as DeepSDF. The authors did not provide more qualitative results in supplementals.

Clarity
This paper is well written and easy to understand. The attached code is well documented and can be deployed.

Conclusion

Overall, this is a well written paper with plausible outcomes. The reviewer believes this paper carries out reasonable efforts and insights into this topic. The reviewer is marginally positive towards its acceptance due to the pleasing results, but is holding a conservative attitude towards its contribution significances. The reviewer would like to see the questions addressed in the rebuttal period, while also refer to other's reviews.

Questions:
1. For each single-view voxel prediction, the paper did not clarify which coordinate system those voxel are in. When aggregating multi-view voxel grid, how is the coordinate transformation handled between different viewpoints? If voxel from different coordinate systems should undertake transformation, how is interpolation handled when merging to a single 32x32x32 grid?
2. Use of GCN. As GCN only optimizes the current mesh, it cannot correct the topology error occurring after the coarse reconstruction. How would this method overcome this, especially when the cubified mesh is in wrong topology?
3. Use of depth. From multi-view predicted depth, one can simply reconstruct from the depths, or run differentiable render for optimizing the mesh geometry directly. Why would we need contrastive depth feature extraction?

---

> ### Author Response · Authors · 2020-11-12
> **Response to AnonReviewer2**
>
> Thank you for the positive review and for taking the time to comment on our paper.
>
> **Q1: The system is rather bulky and ad-hoc**
>
> A: Please refer to the response to Q1 of AnonReviewer5.
>
> **Q2: The paper did not clarify which coordinate system those voxel are in. How is interpolation handled when merging to a single 32x32x32 grid?**
>
> A: As for the coordinate system, we have explained it in the section "Probabilistic Occupancy Grid Merging" of our appendix, we first predict the voxel grids in their respective local coordinate frames, they are then transformed to the coordinate frame of the first view for merging.
> To handle the interpolation, we apply bilinear interpolation on the closest matches in the equivalent global grid from the local grids occupancy values for merging.
>
> **Q3: How would this method recover if the cubified mesh is of wrong topology?**
>
> A: Same as other methods (e.g. Pixel2Mesh, Pixel2Mesh++ and Mesh R-CNN), we can hardly handle the initial mesh with wrong topology. As is shown in Table 1 and Figure 3, our model achieves better quantitative and qualitative results than Pixel2Mesh++. We are happy to regard it as a promising future work.
>
> **Q4: One can simply reconstruct from the depths, or run differentiable render for optimizing the mesh geometry directly. Why would we need contrastive depth feature extraction?**
>
> A: Contrastive depth feature extraction is used to allow the network to reason about required deformation better by contrasting the rendered depths of the current mesh against the predicted depths at different views. As is shown is Table 2, contrastive depth feature extraction increases F1-tau from 79.40% to 80.80% (Row 1), which is a 1.40\% improvement from using only the predicted depth features.
>
> **Q5: The authors did not provide more qualitative results in supplementals.**
>
> A: Sure, we will add more qualitative results in the Appendix B of our revised manuscript.

---

### Official Review · AnonReviewer5 · 2020-11-06
**complicated deep 3D reconstruction pipeline with good results**

**Rating:** 4
**Confidence:** 4

**Review:**


The paper proposes to first predict a coarse (32^3) voxel grid by aggregating independent predictions from individual views. Then, it translate it into a mesh and refine it using deepMVS predictions (using each view in turn as a reference view), and a GCN architecture on the mesh.

On the positive side:
- I like the idea of using MVS-Net, but why not use it from the start (before the single view voxel prediction).
- I think this paper is going toward a render-and-compare approach for 3D shape prediction, which I think is a good idea.
- the boost in the results seems impressive compared to P2M++

There are however several things I don't like or that worry me about this paper:
- the pipeline presented in this paper is extremely complicated, and has many different parts. After reading it, I have no idea what really makes the improvement compared to P2M++. It uses voxels, mesh and depth maps, Graph convolution networks, attention-based architecture, SVR and deepMVS, the training loss has 5 balancing hyperparameters, between things as different as cross-entropy and chamfer distance.
- To me, the ablation studies (Table 2 and 3) show clearly that the most complex parts of the pipeline (3.2, contrastive depth and attention based aggregation) only provides very minor improvements (~1%). Given their complexity and number of hyper parameters, I do not think these can be considered as significative. Given these results, it is completely unclear to me how the proposed approach can lead to a ~14% improvement over pixel2Mesh. I thus think the approach should be strongly simplified (maybe loosing 1% in final performance), but the paper should provide a clear ablation that actually explain why their framework is so much better than P2M++ and this is interesting. Right now, I believe it could be for a bad reason (for example DeepMVS could give excellent results on synthetic data because it is too simple - note I realize that Table 3 shows it is not perfect since there is a further 3.5% boost using GT depth, but it could still be unrealistically good for synthetic data)
- Related work is lacking discussion of important references, namely all classical references for point-based SfM in 2.1 , foldingNet and AtlasNet for mesh generation in 2.2, all implicit volumetric works also in 2,2 (deepSDF, OccupancyNetworks…), the most classical deep depth prediction works in 2.3 (Eigen and Fergus…)

To summarise, despite its impressive numbers, I think this paper cannot be accepted as is, mainly because of its complexity, lack of clear explanation for its huge performance boost, and the only marginal/not significative boosts given by the most complexe parts of the pipeline.

Some additional notes on presentation:
- I am not sure “contrastive depth” is a good choice of name since contrastive feature learning is a popular but unrelated research direction.
- I found 3.2 very hard to parse/re-order. I could only do it with the help of fig. 1 which is itself hard to parse and does not represent e.g. how the attention-based pooling happens

---

> ### Author Response · Authors · 2020-11-11
> **Response to AnonReviewer5**
>
> Thank you for your valuable comments to improve our manuscript.
>
> **Q1: The pipeline presented in this paper is extremely complicated, and has many different parts.**
>
> A: While the pipeline looks a bit complicated, it mainly consists of two parts, similar to Mesh R-CNN. The first part generates a coarse shape, where we extend the ‘Voxel Branch’ in Mesh R-CNN from a single view to multiple views. The second part refines the coarse shape, where we extend the ‘Mesh Refinement Branch’ to multi-view and apply depth images from MVS-Net for better refinement.
> Our losses are similar to Mesh R-CNN for most part. The only new loss term used for 3D reconstruction is the contrastive depth loss. The depth prediction loss is only used to pre-train the depth prediction network (MVS-Net).
> We will improve the writing to make these points clearer.
>
> **Q2: The ablation studies (Table 2 and 3) show clearly that the most complex parts of the pipeline only provides very minor improvements (~1%). It is completely unclear how the proposed approach can lead to a ~14% improvement over Pixel2Mesh++.**
>
> A: We clarify how this 14% improvement is achieved here
> 1. We replace the initial ellipsoid mesh in Pixel2Mesh++ with our coarse shape obtained from single-view voxel prediction. In this way, the F1-tau increases from 66.48\% to 72.74\% (Row 9 in Table 3), which is a 6.26% improvement.
> 2. We replace the single-view voxel prediction with our multi-view voxel prediction. The F1-tau increases from 72.74% to 76.97% (Row 10 in Table 3), which is a 4.23% improvement.
> 3. We replace the multi-view feature pooling with our contrastive depth feature pooling for mesh refinement. The F1-tau increases from 76.97% to 79.63% (Row 5 in Table 3), which is a 2.7% improvement.
> 4. We replace the statistical feature pooling in Pixel2Mesh++ with our proposed multi-head attention feature pooling. The F1-tau increases from 79.63% to 79.82% (Row 1 in Table 3), which is a 0.19% improvement.
> 5. We further apply contrastive depth loss. The F1-tau increase from 79.82% to 80.80% (Row 2 in Table 3), which is a 0.98% improvement.
>
> From the above break down, we can see the reviewer is right that the 4) multi-head attention and 5) contrastive depth loss bring around 1% improvement. The most significant improvements are from 1) using better initial mesh, 2) multi-view voxel prediction, and 3) contrastive depth feature pooling for mesh refinement. All these three components are novel in this method. We will discuss possible simplification of discarding 4) and 5) for a more compact network design and 1% degradation in performance.
>
> **Q3: DeepMVS could give excellent results on synthetic data because it is too simple - note I realize that Table 3 shows it is not perfect.**
>
> A: The quality of predicted depth is not unrealistically good. In fact, the baseline between our input images are arbitrary which is quite challenge for MVS-Net. Furthermore, we do not include the depth refinement component in the original MVS-Net. This is perhaps why the GT depth can further bring a 3.5% improvement. Our source code is attached for verification.
>
> **Q4: Related work is lacking discussion of important references. Section 3.2 needs re-ordering.**
>
> A: Thank you for pointing out those missing references. We will include these point-based SfM papers, implicit volumetric works, and classical deep depth prediction works in the next version and re-arrange Section 3.2 in the revised manuscript.

---

### Author Response · Authors · 2020-11-24
**Rebuttal Revision**

We have uploaded the revised manuscript as per the rebuttal discussions. Following are the changes:
* Rearranged ablation study experiments to more clearly show which component contributes what amount to the final score
* Results when using rendered depths only
* Added more related works on classical references for point-based SfM, foldingNet, AtlasNet, implicit volumetric works, classical deep depth prediction works
* Added appendix B with more qualitative evaluations: best vs pretty models and failure cases
* BerHu loss details and the reason for using it
* Typo corrections

---

### Decision · Program_Chairs · 2021-01-07
**Final Decision**

**Decision:**

Reject

**Comment:**

This submission is an interesting case...

The method it presents appears to work quite well, achieving state-of-the-art quantitative reconstruction results (though qualitatively, the reconstructed surfaces are locally noisy).

The method is quite complex, which different reviewers saw as either a strength or a weakness ("a mix of SoA techniques creatively woven together in a fairly sophisticated model" vs. "bulky and ad hoc").

Most critically: it appears that the reasons for the method's significant (14%) improvement over the prior art for this problem (Pixel2Mesh++) are not due to the novel contributions that the paper focuses on (multi-headed attention, contrastive depth loss). Rather, it is other system design choices that are not novel research contributions that make up all but 1% of this difference (primarily, using a voxel grid predictor to get the initial mesh, as opposed to an initial ellipsoid mesh).

It might be possible for the authors to write a systems paper supporting these design decisions and showing how they lead to better results. However, this is not the paper the authors have written (the majority of the technical detail in the paper is focused on method components that make minimal impact). I would also argue that this hypothetical paper would not necessarily be appropriate for ICLR, since it does not focus on any new representations. It would be better suited to a venue such as CVPR, ICCV, or 3DV.

p.s. Reviewer 5 deserves all of the credit for noticing this major issue with the paper.